# Prevalence and predictors of prenatal depression during the COVID-19 pandemic: A multistage observational study in Beijing, China

**Jin WANG**[1,2], **Libin HU**[1], **Tianyi ZHANG**[1], **Jiajia LIU**[2], **Chuan YU**[2], **Ningxin ZHAO**[3], **Jianlin QI**[2]*, **Lihua LIU**[1]*

1 Institution of Hospital Management, Medical Innovation Research Division of Chinese PLA General Hospital, Beijing, China, 2 Department of Aviation Psychology, Air Force Medical Center, Air Force Medical University, Beijing, China, 3 Faculty of Psychology, Beijing Normal University, Beijing, China

* qjldln888@163.com (JQ); sciaking@sina.com (LL)

**Data Availability Statement:** All relevant data are within the manuscript.

**Funding:** This research received support from the Capital Clinical Application Research Founding of

## Abstract

### Objective

While growing psychological health issues among pregnant women during the COVID-19 pandemic have been clearly validated, most research was conducted in countries with relatively lax quarantine measures. This study aimed to compare the prevalence of prenatal depression among pre-, peak-, and post-COVID-19 in Beijing, the region with a stringent response policy in China. We also explore predictors of prenatal depression throughout the outbreak.

### Methods

We investigated prenatal depression among 742 pregnant women who received antenatal checkups in Beijing from March 28, 2019 to May 07, 2021 using the Edinburgh Postnatal Depression Scale and associative demographic, pregnancy-related, and psychosocial characteristics were measured. The phase was divided into pre-, peak-, and post-COVID-19 in light of the trajectory of COVID-19. Pearson's Chi-square test was used after the examination of confounders homogeneity. The bivariable and multivariable logistic regression was conducted to explore predictors.

### Results

The pooled prevalence of prenatal depression was 11.9% throughout the COVID-19 pandemic. Rates at different phases were 10.6%, 15.2%, and 11.1% respectively and no significant difference was observed. Multivariable logistic regression revealed that history of mental illness, number of boy-preference from both pregnant women and husband's family, social support, occupation, and living space were independent predictors of prenatal depression in Beijing.

Beijing Municipal Science and Technology Committee (Grant number: Z181100001718031). The funder did not involve in any part of the study process, from design to submit the article for publication.

## Conclusion

Our data suggested that the impact of this pandemic on prenatal depression in Beijing appears to be not significant, which will strengthen confidence in adhering to current policy for decision-makers and provide important guidance for the development of major outbreak control and management policies in the future. Our findings may also provide a more efficient measure to identify high-risk pregnant women for professionals and help raise gender equity awareness of pregnant women and their husbands' families. Future studies should focus on the value of targeted care and family relations on the mental health of pregnant women.

## Introduction

Since the initial outbreak in Wuhan, Hubei Province, China, in later December 2019 [1], the novel coronavirus disease 2019 (COVID-19) has rapidly spread around the world [2–4]. World Health Organization (WHO) declared that the global outbreak of COVID-19 was a public health emergency of international concern on January 30, 2020 [5]. This unknown, highly infective, and lethal disease can be considered a global traumatic event and thus amplify the risk of mental illness (e.g. depression, anxiety, posttraumatic stressing disorder) in the general population [6]. A nationally representative survey study in the United States indicated that the prevalence of depression symptoms during COVID-19 was more than 3-fold higher than before [7]. Similarly, relatively high rates of depression symptoms varying from 14.6% to 48.3% are reported in China, Spain, Italy, Iran, the US, Turkey, Nepal, and Denmark during the outbreak [8].

According to the Diagnostic and Statistical Manual of Mental Disorder Fifth Edition (DSM-5), prenatal depression is defined as a major depressive episode (MDD) during pregnancy [9]. It is the most prevalent mental disorder during pregnancy [10] and should be paid more attention during COVID-19. Firstly, women have been found to be more sensitive to stress [11, 12]. Secondly, women during pregnancy may be particularly vulnerable to mental illness during a viral outbreak [13]. The fear of infection and thus the possible severe impact on fetuses and newborns would raise the stress level of pregnant women. As indicated in a special report from the American Journal of Obstetrics and Gynecology, physiological and mechanical changes in pregnancy increase susceptibility to infections, particularly when the cardiorespiratory system is affected, and encourage rapid progression to respiratory failure in the pregnancies, despite no definite evidence of vertical transmission [14]. Moreover, Di Mascio D et al. [15] suggested that COVID-19 infection was associated with a higher rate of preterm birth, preeclampsia, cesarean, and perinatal death in a meta-analysis including 79 hospitalized pregnancies. Likewise, data from Iran reported 7 deaths in 9 pregnant women with severe COVID-19 [16]. Finally, before the infant is exposed to postnatal depression, prenatal depression can independently result in negative developmental consequences [17] and has been demonstrated to expert significantly detrimental effects on pregnant women and the developing fetus, even extending into childhood and possibly adulthood [18]. As mentioned above, women, especially pregnant women, should be given special regard during COVID-19 to protect them from prenatal depression.

Prenatal depression during the pandemic has been extensively studied, but no consensus reached. For example, by separating healthy, asymptomatic individuals exposed to infectious

diseases during the incubation period, quarantine can effectively contain the spread of the disease [19]. However, rigid quarantine procedures including social distancing and isolation increased the risk of being exposed to psychological illness among pregnant women [20]. Besides, the paused prenatal care, passive lifestyle, and job-losing brought by quarantine exacerbated the pregnancy-related mental health issues [21, 22]. In contrast, Chinazzi et al. [23] found that Chinese strict travel restrictions delayed the overall epidemic progression and had a positive effect on the reduction of international case importations, thus alleviating the fear of infection among pregnant women presumably. Therefore, it is necessary to investigate prenatal depression throughout the pandemic in China.

While several literatures have investigated prenatal depression by comparing the prevalence of prenatal depression among different phases of COVID-19 in countries with relatively lax quarantine strategies, like the US, Italian, and Canada [24–26], this method has been rarely applied in China. Hu et al. [27] conducted a cross-sectional study during COVID-19 in 3 Chinese cities (including Beijing), revealing the prevalence and relevant factors of prenatal depression in different stages. However, the impact of COVID-19 on prenatal depression was not conclusive due to the lack of samples before the pandemic. Similarly, although pre-[28] and post-COVID-19 [29] studies have been reported separately, no comparison among the different stages was performed. Additionally, these studies failed to involve gender preference, which is closely related to prenatal depression in low- and middle-income countries [30].

Furthermore, it is important to consider potential demographic predictors when identifying high-risk pregnant women for timely intervention for prenatal depression. Pregnancy can be a challenging time for women, with physical and biological changes that can increase the risk of depression [31]. Factors such as gestational week, number of pregnancies, history of abortion, pregnancy intentions, and sex preference can also contribute to prenatal depression. In some cultures, such as China, mothers and their husbands' families may prefer boys, which can further exacerbate the risk of prenatal depression [32]. A history of mental illness, including anxiety, depression, or psychiatric treatment at any point in a woman's life, is a well-known risk factor for prenatal depression [33]. Additionally, the strong association between poor social support and higher risk of prenatal depression has been validated in a causal modeling study [34].

To this end, this multistage, single-center, observational study was performed to analyze the prevalence of prenatal depression in Beijing, a stringent quarantine region in China, throughout the outbreak and to determine the demographic, pregnancy-related, and psychosocial predictors. This study will clarify the effects of a drastic quarantine policy on prenatal depression for policymakers and provide a guideline for psychiatrists and obstetrics on prevention and intervention.

## Methods

### Participants

Participants in this study were pregnant women who received routine prenatal checkups (ANC) from March 28, 2019 to May 07, 2021 at a hospital located in the Haidian district, Beijing, which was appointed as a prenatal psychological care service by the Beijing Municipal Health and Family Planning Commission [35]. This study was approved by the ethical committee of the Air Force Medical Center. Inclusion criteria for eligibility were pregnant women (2–9 months of pregnancy) who were aged 20 years or older, Chinese-speaking, confirmed their written consent, and were in the absence of a current life-threatening illness or severe psychiatric disorder (e.g., bipolar disorder, schizophrenia, or neurodevelopmental disorder).

## Procedures

Participants completed 2 standardized measures under the guidance of 2 psychology postgraduates and were interviewed by 2 nurses using a structured questionnaire including demographic, pregnancy-related, and psychosocial assessments (described below). 4 field researchers had no prior established relationship with the participants and received 2 days of training including quality control, instruction of questionnaires, completeness of the information, and research ethics. Data were coded, cleaned, edited, and entered into EpiData version 3.1 to minimize logical errors. Participants who completed the study entered into a prize draw for 1–10 RMB. Women with a high probability of prenatal depression, regardless of inclusion or not in this study, were directed to contact professional healthcare.

## Measures

**Phases of the COVID-19 pandemic.**   In light of announcements and data from the Chinese National Health Commission (NHC), Beijing Health Commission (BHC), and China Central Television (CCTV, the most authoritative media in China), phases were cut as *pre-COVID-19*, *peak-COVID-19, and post-COVID-19* by the trajectory of the outbreak in Beijing. Specifically, January 20, 2020, when Chinese top respiratory expert Zhong Nanshan indicated that the virus could spread from person to person on CCTV [36], was labeled as the initiation of peak-COVID-19, and August 25, 2020, as the termination of peak-COVID-19, when BHC [37] formally informed that all confirmed cases in Beijing have been cleared in Xinfadi, Beijing. Consequently, the duration of peak-COVID-19 in this study was 218 days. The pre-COVID-19 and post-COVID-19 were respectively defined as March 28, 2019-November 01, 2019 and October 01, 2020-May 07, 2021 considering the following reasons: 1) equal days of duration; 2) joint consideration of time effect and the first reported case in December 2019 (pre-COVID-19); 3) elimination of residual effects (post-COVID-19).

**Depression assessment.**   The 10-item Edinburgh Postnatal Depression Scale (EPDS) is a widely implemented self-report questionnaire assessing symptoms of depression. It was measured on a 4-point scale, yielding total scores ranging from 0 to 30 where higher scores indicate more severe symptoms [38]. Participants were instructed to answer the questions based on their last 7 days experience. Though originally developed to detect postpartum depression, EPDS has been validated for good to acceptable sensitivity in the prenatal period across the world [39]. The Mainland Chinese version of EPDS has high sensitivity and specificity for detecting depression in pregnant women at a cut-off score of >10 in several Chinese validation studies [40, 41].

**Demographic assessment.**   Demographic characteristics included maternal age, education attainment, occupation, per capita living space ($m^2$), per capita monthly household incomes (RMB), and co-living at the time of pregnancy. Maternal age was categorized as 20–25, 26–30, 31–35, and > 35 years. Educational attainment included college degree or below, bachelor degree, Master degree or above. The occupation was classified as employed and unemployed. Per capita living space was defined as family living space divided by the number of adults and children and 21 $m^2$ was signed as the cut-off point of small. The calculation formula of the per capita monthly household income was the average monthly household income in the previous year divided by the number of adults and children. Participants made a selection from ≤5000 and >5000, which was defined as the poverty line according to data from the Beijing statistical yearbook, Beijing Bureau of Statistics [42]. Co-living in pregnancy was segmented as alone, husband only, husband's family, and others.

**Pregnancy-related assessment.**   Individual body mass index (BMI; $kg/m^2$) was calculated as weight in kilograms divided by height in meters squared during ANC and adhered to the Chinese classification of underweight (<18.5 $kg/m^2$), normal weight (18.5–23.9 $kg/m^2$),

overweight (24–27.9 kg/m$^2$) and obese ($\geq$28 kg/m$^2$) [43]. Gravidity included primi-gravida (first pregnancy) and multi-gravida ($\geq$2 pregnancy experiences), while the history of abortion (Yes, No) covered both the induced and spontaneous ones. Measured by the date of the last menstrual period and ultrasound assessment, Gestational weeks was categorized as the first ($\leq$12 weeks), second (13–26), and third ($>$ 26 weeks) trimester. Other self-reported characteristics were pregnancy intentions (planned conception, unplanned pregnancy), expected mode of delivery (vaginal delivery, cesarean delivery, not considered), perception of family care (satisfied, dissatisfied, medium), and sex preference of pregnant women and husband's family (boy, girl, not considered). To further investigate the association between sex preference and prenatal depression, we constructed a composite variable by the number of boy-preference for analysis: 0, 1, and 2 (0 = neither pregnant women nor their husbands' families have boy-preference; 1 = one part has boy-preference, 2 = both have boy-preference).

**Psychosocial assessment.** Psychosocial characteristics comprised the history of mental illness, which is labeled as a history of a diagnosed mental disorder prior to pregnancy, and social support. Social support was measured by the Chinese version of the Social Support Rating Scale (SSRS), which was originally developed by Xiao Shuiyuan in 1986. It has been widely used in various populations and shown good validity and reliability [44, 45]. The SSRS consists of 10 items mostly scored on a 4-point Likert scale, The total scores ranging from 12 to 66 are used to assess the current level of social support. Based on a previous study, total scores were defined as low $\leq$ 44 and high (45–66) [46].

## Statistical analysis

The detailed data are available in S1 File of this research. Statistical analyses were performed using the IBM Statistical Package for the Social Sciences (SPSS Version 24 for Windows). All data were categorical variables and presented as absolute numbers and percentages (%). To identify the prevalence variation of prenatal depression among 3 phases of COVID-19, Pearson's Chi-square test was used after the examination of confounders homogeneity. The bivariable and multivariable logistic regression were conducted to identify the independent predictors. The binary form of the dependent variable was coded as "1" for prenatal depression (EPDS score$>$10) and "0" for the absence (EPDS score $\leq$ 10). All variables with a p-value of 0.2 or lower in bivariable analysis were selected as potential predictors for the multivariable logistic regression model according to the suggestion of Hosmer and Lemeshow [47], crude odds ratios (cORs) and 95% confidence intervals (CIs) were calculated. Subsequently, a multivariable logistic regression model with a stepwise procedure was established, in which the adjusting odds ratios (aORs) and 95% CIs were estimated as indicators of the strength of association, and those variables with $p$ <0.05 were considered to be statistically significant. The Hosmer-Lemeshow goodness of fit test was used to assess the fitness of this model. All statistical tests were two-sided. In addition, to determine the probability of detecting an effect of a given size with a given level of confidence under constraints of the sample size, we also conducted a post-hoc power analysis in G Power 3.1.9.4 [48]. This analysis suggested that the sample size provided sufficient statistical power for each statistically significant predictor ($>$0.90).

## Results

### Descriptive statistics

A total of 977 pregnant women received ANC from March 28, 2019 to May 07, 2021. Among them, 187 (19.1%) who rejected consent were excluded and the response rate was 81.9%. 48 (4.9%) with missing data were deleted. Consequently, 742 participants were included in the final sample. Table 1 presents the overall characteristics of the participants. More than half of

**Table 1. Characteristics of pregnant women throughout the COVID-19 in Beijing, China.**

| Variables | Frequency | Percentage (%) |
|---|---|---|
| Age (years) | | |
| 20–25 | 9 | 1.2 |
| 26–30 | 213 | 28.7 |
| 31–35 | 374 | 50.4 |
| >35 | 146 | 19.7 |
| BMI[a] | | |
| underweight | 49 | 6.6 |
| normal weight | 506 | 68.2 |
| overweight | 155 | 20.9 |
| obese | 32 | 4.3 |
| Education attainment | | |
| college degree or below | 63 | 8.5 |
| bachelor degree | 317 | 42.7 |
| master degree or above | 362 | 48.8 |
| Occupation | | |
| employed | 712 | 96.0 |
| unemployed | 30 | 4.0 |
| History of mental illness[b] | | |
| yes | 12 | 1.6 |
| no | 730 | 98.4 |
| Co-living | | |
| husband only | 402 | 54.2 |
| alone | 53 | 7.1 |
| husband's family | 134 | 18.1 |
| others | 153 | 20.6 |
| Per capita monthly household incomes[c] | | |
| non-poverty | 684 | 92.2 |
| poverty | 58 | 7.8 |
| Per capita living space[d] | | |
| non-small | 498 | 67.1 |
| Small | 244 | 32.9 |
| Gestational weeks[e] | | |
| first | 179 | 24.1 |
| second | 413 | 55.7 |
| third | 150 | 20.2 |
| Gravidity[f] | | |
| primi-gravida | 363 | 48.9 |
| multi-gravida | 379 | 51.1 |
| History of abortion[g] | | |
| yes | 106 | 14.3 |
| no | 636 | 85.7 |
| Pregnancy intentions | | |
| planned conception | 489 | 65.9 |
| unplanned pregnancy | 253 | 34.1 |
| Perception of family care | | |
| satisfied | 675 | 91.0 |
| medium | 39 | 5.3 |

(*Continued*)

**Table 1.** (Continued)

| Variables | Frequency | Percentage (%) |
|---|---|---|
| dissatisfied | 28 | 3.8 |
| Expected mode of delivery | | |
| vaginal delivery | 448 | 60.4 |
| cesarean delivery | 84 | 11.3 |
| not considered | 210 | 28.3 |
| Boy-preference[h] | | |
| 0 | 652 | 87.9 |
| 1 | 72 | 9.7 |
| 2 | 18 | 2.4 |
| Social support | | |
| high | 430 | 58.0 |
| low | 312 | 42.0 |
| Phases of the COVID-19[i] | | |
| pre- | 310 | 41.8 |
| peak- | 171 | 23.0 |
| post- | 261 | 35.2 |

[a]Calculated as weight in kilograms divided by height in meters squared; stratified as underweight ($<18.5$ kg/m$^2$), normal weight (18.5–23.9 kg/m$^2$), overweight (24–27.9 kg/m$^2$), and obese ($\geq 28$ kg/m$^2$).

[b]Had a history of a diagnosed mental disorder prior to pregnancy.

[c]Calculation formula was the average monthly household income in the previous year divided by the number of adults and children in their house. 5000 RMB was defined as the poverty line according to data from the Beijing statistical yearbook, Beijing Bureau of Statistics.

[d]Defined as family living space divided by numbers of adults and children and 21 m$^2$ was signed as the cut-off point of small size.

[e]Categorized as the first ($\leq 12$ weeks of gestation), second (13–26 weeks of gestation), and third ($>26$ weeks of gestation) trimester.

[f]Primi-gravida (first pregnancy) and multi-gravida ($\geq 2$ pregnancy experiences).

[g]Covered both induced and spontaneous abortion.

[h]Categorized by the number of boy-preference among husband's family and pregnant women.

[i]Pre-COVID-19 (March 28, 2019-November 01, 2019), peak-COVID-19 (January 20, 2020- August 25, 2020), and post-COVID-19 (October 01, 2020-May 07, 2021).

the pregnant women (50.4%) were 31–35 years old, 68.2% were normally weight, nearly half (48.8%) had a master degree or above, most (96.0%) were employed and 12 (1.6%) had a history of mental illness. Those of low per capita monthly household incomes and living space were 7.8% and 32.9% respectively. More than one-third (34.0%) were unplanned pregnancy, only 3.8% were dissatisfied with family care, and 87.9% reported an absence of boy-preference from both husband's family and themselves. Lastly, 41.8%, 23.0%, and 35.2% received ANC at pre-, peak-, and post-COVID-19 respectively.

## Prevalence of prenatal depression

As shown in Fig 1, the percentage of participants who scored on the EPDS $>10$ was 10.6%, 15.2%, and 11.1% at pre-, peak-, and post-COVID-19 respectively, and the pooled prevalence was 11.9%. Phases of COVID-19 presented homogeneous demographic, pregnancy-related, and psychosocial characteristics (S1 Table). There is a small prevalence variation throughout

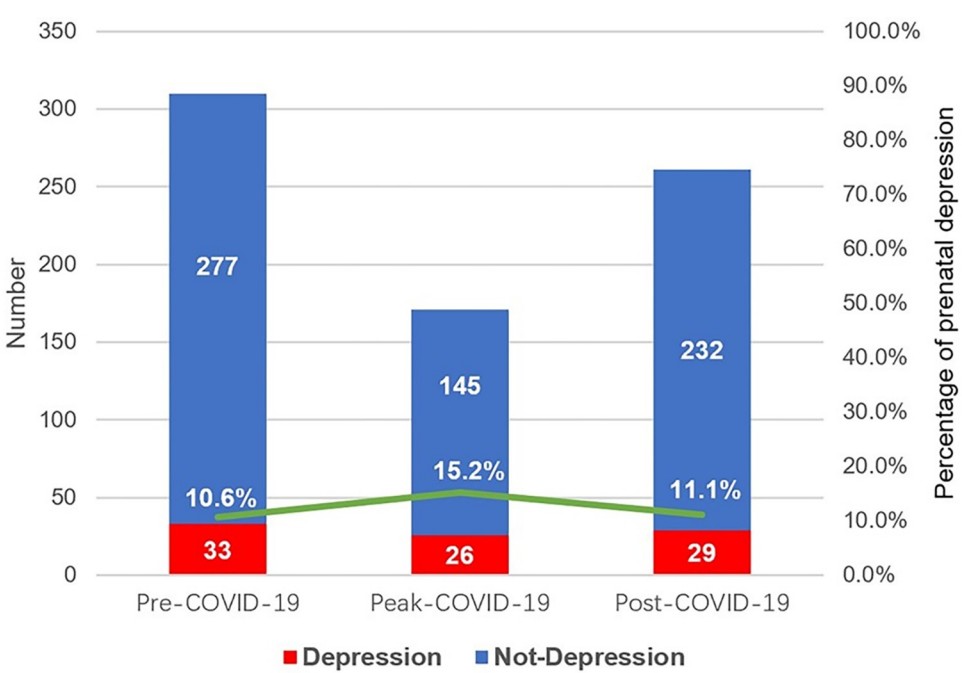

**Fig 1. Prenatal depression at pre-, peak-, and post-COVID-19 in Beijing, China.** The numbers in the blue and red bars indicated the number of pregnant women without and with depression, and the percentages represented the prevalence of prenatal depression at the corresponding phase.

the pandemic in Beijing, China, and Pearson Chi-square revealed no significant difference among phases ($\chi^2 = 2.408$, $p = 0.300$).

## Bivariable analysis

Fig 2 illustrated the relationship between variables and prenatal depression in the bivariable model. Among the demographic characteristics, the prevalence decreased with education attainment ($p<0.05$). An elevated rate of prenatal depression was observed in unemployed women compared to employed ones ($p<0.01$). Other demographic characteristics including age, co-living, per capita monthly household income, and per capita living space showed $p$ values higher than 0.05.

Regarding pregnancy-related characteristics, the bivariable analysis showed 3 variables with a $p$ value lower than 0.05: pregnancy intentions ($p = 0.005$), perception of family care ($p = 0.001$), and boy-preference ($p<0.001$). Variables including BMI, gestational weeks, gravidity, history of abortion, and expected mode of delivery showed $p>0.05$ by bivariable analysis. Concerning psychosocial characteristics, both history of mental illness and social support ($p<0.001$) were significantly associated with prenatal depression.

## Multivariable analysis

Fig 3 demonstrated the predictors of prenatal depression during the COVID-19 pandemic in a fully adjusted model. Multivariable analysis of demographic, pregnancy-related and psychosocial characteristics with $p$ values equal to or lower than 0.2 by bivariable analysis revealed that history of mental illness (aOR = 9.55; 95% CI: 2.57–35.51; $p = 0.001$), number of boy-preference (aOR = 6.91; 95% CI: 2.31–20.64; $p = 0.001$), social support (aOR = 3.17; 95% CI: 1.88–5.35; $p<0.001$), occupation (aOR = 2.73; 95% CI: 1.12–6.70; $p = 0.028$), and per capita living

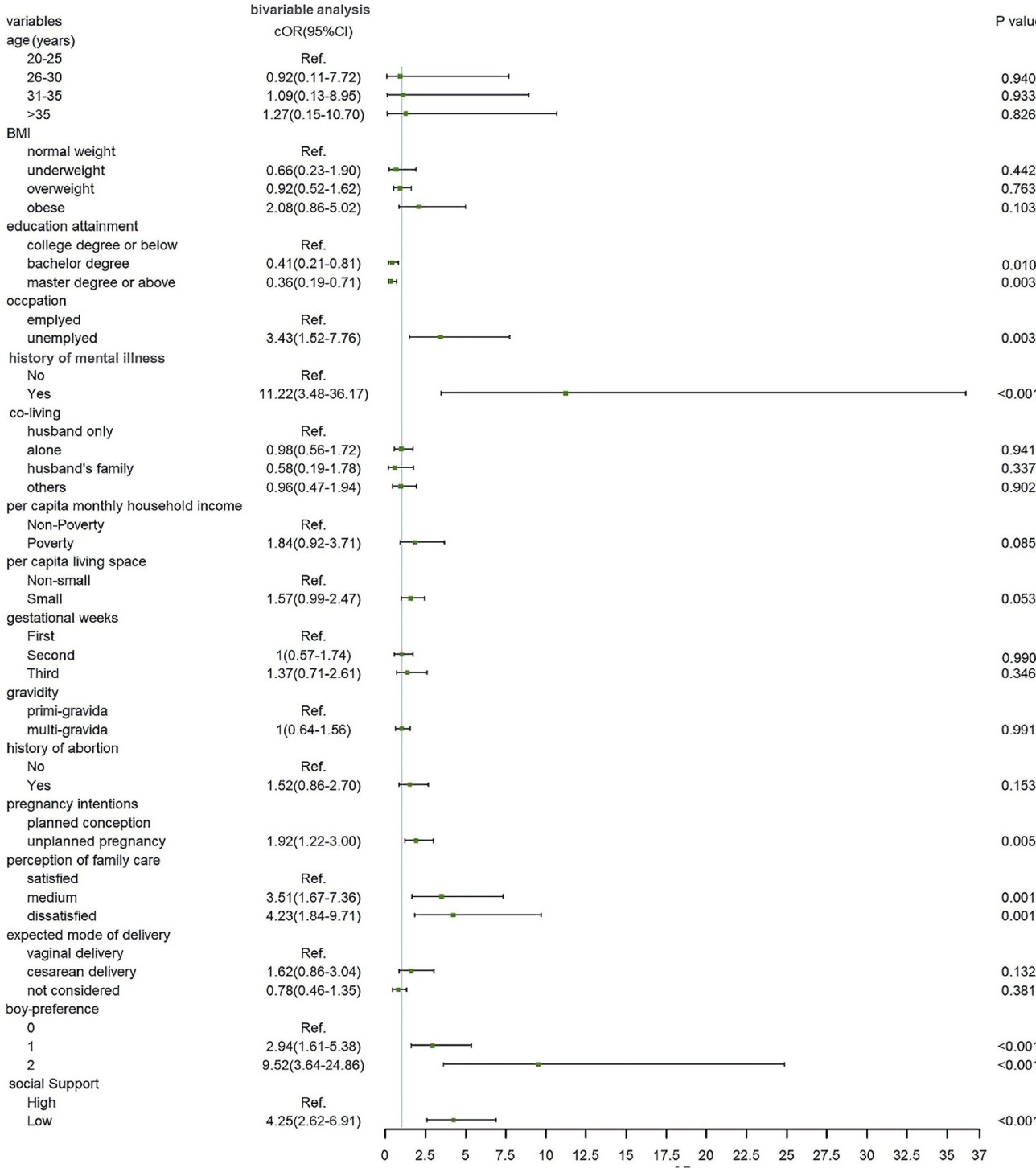

**Fig 2. Crude association between characteristics and prenatal depression in the bivariable model.**

space (aOR = 1.77; 95% CI: 1.06–2.95; $p$ = 0.030) were significant predictors of prenatal depression. Besides, a dose-response relationship existed between the number of boy-preference and prenatal depression. The result of the Hosmer-Lemeshow test ($p$ = 0.340) indicated

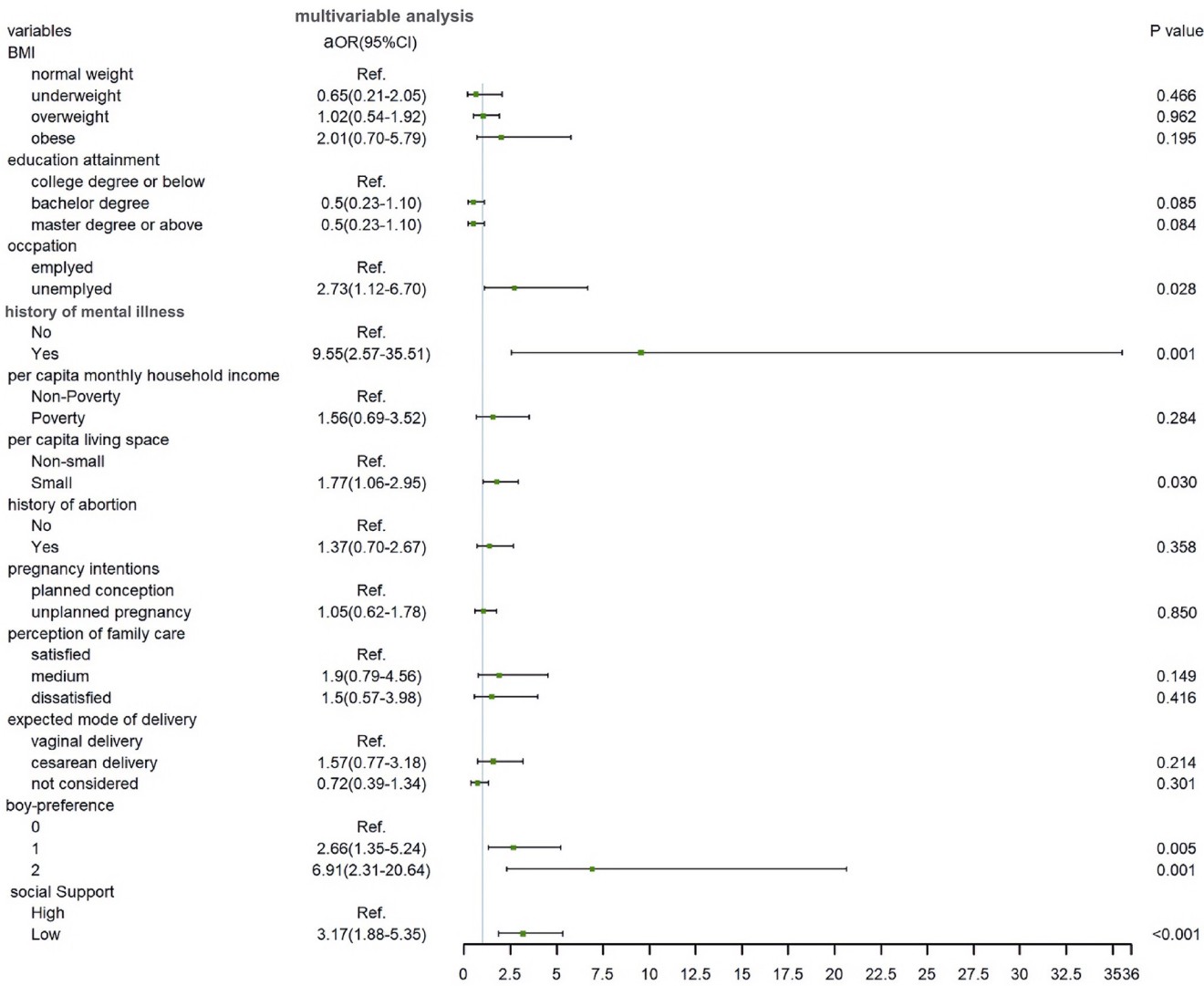

**Fig 3. Adjusted odds ratio of prenatal depression during the COVID-19 pandemic in multivariable regression model.**

an acceptable fit of our multivariable regression model. Nagelkerke's pseudo $R^2$ of 0.231 indicated that the significant variables explained 23.1% of the variance.

## Discussion

Overall, our findings showed that the prevalence of prenatal depression at pre-, peak-, and post-COVID-19 was 10.6%, 15.2%, and 11.1% respectively, and the pooled rate was 11.9% in Beijing, China. Despite a small variation, there was no significant difference in prenatal depression among 3 phases. The history of mental illness, number of boy-preference, low social support, unemployed, and small living space were independent predictors of prenatal depression. In addition, number of boy-preference from both pregnant women and their husbands' families had a dose-response relationship with prenatal depression. These predictors remained significant even controlled for age, BMI, education attainment, co-living, per capita monthly household income, gestational weeks, gravidity, history of abortion, pregnancy intention, perception of family care, and expected mode of delivery.

## Prevalence throughout the COVID-19 pandemic

This study showed that the prevalence of prenatal depression in Beijing was 10.6% before the outbreak, which was in the range from 5.4% to 37.4% reported in China, the US, Mexico, Spain, Italy, and Ethiopia [30, 49–55], but lower than the pooled prevalence estimate of 25.3% in a meta-review including 3 studies in China [10]. One account for the difference may be the sample characteristics. Participants in this study were recruited from theHaidian district, which is the educational and cultural center of China. Higher socioeconomic status and education attainment potentially protects pregnant women from mental health issues presumably [56]. Besides, this variation may be derived from the diverse tools, gestation, and culture.

It is noticeable that a significant gap existed in the prevalence of prenatal depression at the peak of the outbreak in our study compared with results in similar studies from other regions. A previous finding of King LS et al. [57] in the US indicated that pregnant women during COVID-19 were approximately twice likely to have depression than matched women before the pandemic, 25.0% and 51.0% respectively. Recently, a cross-sectional study in the UK reported the pandemic has increased the prevalence of depression to 47% among expectant mothers [58]. Broadly distinct epidemic response strategies probably contributed largely to this diverse outcome. Since the first case of COVID-19 was reported on Jan 20, 2020 [59], the US declared suspension of unessential businesses and schooling at the state and local levels. However, such a temporary strategy and separate administration generated uncertain outbreaks in both space and time [60], especially confronted with the continued increase in identified cases and the constant emergence of new strains. Moreover, pregnant women may be forced to choose between job loss and infection risk for a lack of paid parental leave in the US [61]. The UK government planned to achieve herd immunity from the outset, whereby civilians would gain immunity from the Coronavirus through widespread exposure [62]. Similar measures would lead to public panic and uncertainty [63] and thus increase prenatal depression.

In contrast, the Chinese government has been persistently implementing the dynamic zero strategies, which implies rapid extinguishment of the outbreak once native cases appear through integrated prevention and control measures, including proactive screening, outbreak sites control, close contacts management, crowd aggregation reduction, and rapid and maximum treatment [64]. These nationally drastic control policies effectively mitigated the spread of the disease [65] and thus decrease the risk of infection in pregnant women. Therefore, it is reasonable to infer that despite the negative effect of social interaction, this stringent and effective epidemic prevention policy reduces the level of stress to COVID-19 in pregnant women, leading to an absence of significant variation in the prevalence of prenatal depression among the pre-, peak-, and post-COVID-19 in Beijing, China. This will shed light on the development of major outbreak control and management policies in the future.

## Predictors of prenatal depression

Align with previous studies [49, 51], a history of mental illness was the strongest predictor of prenatal depression in our findings. Women with a history of mental health issues have an approximately 9-fold increased risk of depression during pregnancy.[33]. Our data showed that compared with the situation neither pregnant women nor husband's family has boy-preference, when one part has boy-preference, the risk of depression for pregnant women was 2.66 higher and increased to 6.91 when pregnant women were exposed to dual pressure. Previous studies have investigated the sex preference of pregnant women and their husbands' families respectively. Joshi et al. [30] showed a close association between sex preference, particularly boy-preference, and prenatal depression in low- and middle-income countries. However, to

our knowledge, no other studies have combined these two variables. Therefore, we created a new composite variable, the number of boy-preference, by recoding the preference from pregnant women and their husbands' families to explore the unique importance of their combination. Our data suggested that pregnant women who suffered from both internal and external pressure from themselves and their husbands' families were at higher risk of prenatal depression. Furthermore, our finding highlighted the influence of deep-rooted social belief of lineage perpetuation generated by the husband's family on women [66].

We also found that low social support was an independent predictor of prenatal depression during the outbreak, in alignment with studies before COVID-19 [67]. Findings showed that social isolation generated by quarantine (e.g., lockdown, travel restriction) reduced levels of apparent social support from families and friends [68], and lack of social support further led to loneliness and repetitive negative thinking [69]. Hence, sufficient social support was a powerful protector for prenatal depression, especially during the COVID-19 pandemic. Besides, unemployed and small living spaces were associated with increased odds of prenatal depression. These two predictors led women to face economic and circumstance pressure which further aggravated prenatal depression symptoms during the epidemic [70].

Notably, several characteristics validated in past studies appeared unrelated to prenatal depression in the multivariable analysis, including education, unplanned pregnancy, and perception of family care, despite significant association observed in the bivariable analysis. The complex interplay between the variables and social support contributed to these results. Higher social support can reduce perceived stress and play a moderating role in prenatal depression symptoms [71]. On the other hand, due to sample design, the distribution of variables in this study was quite distinct from previously-conducted ones.

## Strengths and limitations

This study may be the first to investigate prenatal depression throughout the COVID-19 pandemic in Beijing, China. By constraining samples selected within a single center, confounders homogeneity was ensured among different phases of the pandemic. Besides, the composite variable calculated according to sex preference from both pregnant women and their husbands' families further revealed the close association between boy-preference and prenatal depression. It is important to consider the limitations of the current study. Firstly, certain potential predictors, including perceived stress, stressful life events, substance history, and history of medical illness, as well as pandemic-related variables were not included in this study. Secondly, due to the limited window for prenatal depression measurement and the uncertain of the COVID-19 development trajectory, employing the same individuals among three phases is difficult to achieve. Therefore, the selection bias should be concerned, although the homogeneity of confounding variables has been assessed. Thirdly, no further diagnosis was performed after screening for depression symptoms. Thus, the prevalence of prenatal depression should be generalized prudently among a larger population. Nationally representative study studies are needed in the future for a detailed clarification of how epidemic prevention policy affects mental health among pregnant women.

## Conclusion

We concluded that the public and professionals should take prenatal depression seriously during the COVID-19. Our findings suggest that the impact of this pandemic on prenatal depression in Beijing appears to be not significant, in light of the strict, coherent, and effective strategies implemented, which will strengthen confidence in adhering to current policy for decision-makers and provide important guidance for the development of major outbreak

control and management policies in the future. We also found that history of mental illness, number of boy-preference from both pregnant women and husband's family, social support, occupation, and living space were significant predictors of prenatal depression. This implies the necessity of integrating demographic, pregnancy-related, and psychosocial factors into prenatal depression screening and may provide a more efficient measure to identify high-risk pregnant women for professionals. Furthermore, the study helps raise gender equality awareness of both pregnant women and their husbands' families. Future studies should focus on the value of targeted care and family relations on the mental health of pregnant women.

## Supporting information

**S1 Table. Homogeneity test of confounders among different phases of COVID-19 in Beijing, China.**
(DOCX)

**S1 File. Original dataset.**
(RAR)

## Acknowledgments

The authors thank all the participants for their time and efforts. The authors are grateful to advanced students (Yuejiao HOU, Xichao WANG) and nurses (Xiaomei ZHOU, Nan LI) in this study for their support in the data collection.

## Author Contributions

**Conceptualization:** Jianlin QI, Lihua LIU.

**Data curation:** Jiajia LIU.

**Formal analysis:** Jin WANG.

**Funding acquisition:** Jianlin QI.

**Investigation:** Libin HU, Jiajia LIU, Chuan YU.

**Methodology:** Tianyi ZHANG, Chuan YU, Ningxin ZHAO.

**Project administration:** Jianlin QI.

**Software:** Ningxin ZHAO.

**Supervision:** Lihua LIU.

**Validation:** Lihua LIU.

**Visualization:** Libin HU, Tianyi ZHANG.

**Writing – original draft:** Jin WANG.

**Writing – review & editing:** Jin WANG, Ningxin ZHAO.

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
