## [Decision Letter · Decision Letter 0]

4 Jul 2023

PONE-D-22-21013Prevalence and Predictors of Prenatal Depression during the COVID-19 Pandemic: A Multistage Observational Study in Beijing, ChinaPLOS ONE

Dear Dr. LIU,

Thank you for submitting your manuscript to PLOS ONE. After careful consideration, we feel that it has merit but does not fully meet PLOS ONE’s publication criteria as it currently stands. Therefore, we invite you to submit a revised version of the manuscript that addresses the points raised during the review process.

Please address all comments raised by the reviewers. ==============================

We look forward to receiving your revised manuscript.

Kind regards,

Abel Fekadu Dadi, Ph.D.

Academic Editor

PLOS ONE

Journal Requirements:

“This research received support from the Capital Clinical Application Research Founding of Beijing Municipal Science and Technology Committee (Grant number: Z181100001718031). The funder did not involve in any part of the study process, from design to submit the article for publication.”

3. PLOS requires an ORCID iD for the corresponding author in Editorial Manager on papers submitted after December 6th, 2016. Please ensure that you have an ORCID iD and that it is validated in Editorial Manager. To do this, go to ‘Update my Information’ (in the upper left-hand corner of the main menu), and click on the Fetch/Validate link next to the ORCID field. This will take you to the ORCID site and allow you to create a new iD or authenticate a pre-existing iD in Editorial Manager. Please see the following video for instructions on linking an ORCID iD to your Editorial Manager account: https://www.youtube.com/watch?v=_xcclfuvtxQ.

“This research received support from the Capital Clinical Application Research Founding of Beijing Municipal Science and Technology Committee (Grant number: Z181100001718031). The funder did not involve in any part of the study process, from design to submit the article for publication.”

“This research received support from the Capital Clinical Application Research Founding of Beijing Municipal Science and Technology Committee (Grant number: Z181100001718031). The funder did not involve in any part of the study process, from design to submit the article for publication.”

Additional Editor Comments (if provided):

Please address the comments by the reviewers.

Reviewers' comments:

Reviewer's Responses to Questions

**Comments to the Author**

1. Is the manuscript technically sound, and do the data support the conclusions?

Reviewer #1: Yes

Reviewer #2: No

2. Has the statistical analysis been performed appropriately and rigorously? 

Reviewer #1: Yes

Reviewer #2: No

3. Have the authors made all data underlying the findings in their manuscript fully available?

Reviewer #1: Yes

Reviewer #2: Yes

4. Is the manuscript presented in an intelligible fashion and written in standard English?

Reviewer #1: Yes

Reviewer #2: No

5. Review Comments to the Author

Reviewer #1: Dear Authors,

Thank you for considering this journal. I found your work as invaluable research that may close the gap and add to the literature. I have some concern about methodology of the study though. Please address my concerns in your revision and resubmit for further review.

1. My first concern is about the risk of bias in collecting data and reporting results in this study. Authors measured depression (interviews/survey) in different individuals in different phases of Covid 19 outbreak. We are aware that depression depends on various factors like socio-demographic characteristics, economic status, age, number of pregnancies, and mental status at the time of the study (as the authors acknowledged and discussed in the method section). However, I am still uncertain how authors reduced the risk of bias in reporting the prevalence in different phases of Covid 19 outbreak. I mean, how did they relate a higher rate of depression in certain individuals who were surveyed during the peak phase to the outbreak? For example, those who attended the study during the peak phase are not the same individuals who did so before or post the peak phase. I like to see what strategies the authors applied to reduce the risk of bias in reporting results.

2. Another concern is about collecting data for the pre-peak phase of covid 19. How did the authors collect data? Did you conduct the study from March 2019 to May 2021, then classified the collected data into 3 phases (after completing data collection)?! Is that right? I mean, how did the authors predict there would be a peak phase for Covid 19? I would like to see an explanation about this.

Reviewer #2: Comments and suggestions to the authors

In the first place I would like to thank the academic editor for inviting me to review this manuscript.

Studying prevalence and Predictors of Prenatal Depression during the COVID-19 Pandemic has paramount clinical and public health significance. However, the work has to be reviewed promptly before used by the scientific community. Thus, the general comments, suggestions and questions have been attached as follows.

1. The body of knowledge that your study added

What was the necessity for your research, given that there have already been studies on this topic in many regions of China that have taken into account essential factors that you overlooked, and did you feel that your research contributed anything new to the body of knowledge? (I have attached some of similar works conducted previously).

I noticed that there is a duplication of efforts.

• https://www.frontiersin.org/articles/10.3389/fpsyt.2023.1191152/full

• https://bmcpregnancychildbirth.biomedcentral.com/articles/10.1186/s12884-022-04428-1

• https://bmjopen.bmj.com/content/10/9/e038511

Moreover, the data was taken from 2019-2021, and it would have been interesting if it was accessible for the scientific community during the rapid raise of the covid-19 pandemic. However, it is too late to be used by the scientific community (two years later than the time of data collection).

2. Abstract

The abstract did not adhere to the PLOS ONE’s manuscript submission guidelines:

The abstract should describe the main objective(s) of the study, the methods, the results and the conclusion. The subheading “Purpose” is not included in the PLOS ONE Manuscript Submission Guidelines.

3. Introduction

You stated ‘’Similarly, relatively high rates of symptoms of depression varying from 14.6% to 48.3% are reported in China, Spain, Italy, Iran, the US, Turkey, Nepal, and Denmark during the outbreak [8)’’.However, you missed the citation of all the references for studies conducted in various countries which actually indicated poor referencing for your evidences.

The gaps in the research published so far have not been adequately explained, nor the significance of your study compared to previous studies. The introduction also lacks focus to the study target (prenatal depression and its predicators). Moreover, the magnitude and important predictors of prenatal depression during the pandemic are not clearly stated. Overall, it needs general improvement.

4. Methods

Ethical issues

The eligibility criterion lacks clarity.

Inclusion criteria for eligible were pregnant women (2-9 months of pregnancy) who were aged 20 years and older, Chinese-speaking, confirmed their written consent, and had no plans to leave residence within 1 year. Women who had life-threatening illness or severe psychiatric disorders (e.g., bipolar disorder, schizophrenia or neurodevelopmental disorder) were excluded.

• Questions

Is it ethical to exclude non -Chinese speaking pregnant mothers, and planned to leave before one your but suffering from severe prenatal depression?. What about those mothers with severe depression. Because, you stated that “Women with a high probability of prenatal depression were directed to contact free professional healthcare. How did you see the deliberate exclusion of mothers with depression from health care service based on their language from the ethical perspective? Why you included mothers >20years of age. What if mothers at the age of 18, 19?

Assessment tool

• Psychological assessment tool

You stated “Psychosocial characteristics comprised mental history and social support’’. Here, the word mental history is a vague and stigmatizing word, it is better to replace with’ history of mental illness”

Based on a previous study, total scores were defined as low (< 44) and high (45-66). What if the score becomes exactly 44???

Statistical analysis

• This sentence is ambiguous for the reader.

All data were categorized variables and were presented as absolute number and percentage (%). To identify the association between prevalence and phases of COVID-19, Pearson’s Chi-square was used after the examination of confounder’s homogeneity.

All variables with a p-value of 0.2 or lower in bivariate analysis were selected as potential predictors for the multivariate logistic regression model according to the suggestion of Hosmer and Lemeshow [37], crude odds ratios (cORs) and 95% confidence intervals (CIs) were calculated.

• Question

Why only variables with a p-value <=0.2 in the bivariate analysis were taken in to multi-variate analysis. It is generally recommended to included variables known to affect the dependent variable (prenatal depression in your case) in various studies rather than taking variables with a p-value<=0.2 in the bivariable analysis.

• Operational definition

History of mental illness was not clearly operationalized and in Table 1 you defined mental history as “b Present depression, anxiety or psychosomatic symptoms prior to pregnancy” which is to mean history of common mental disorder. What about mothers with history of other mental health problems such as schizophrenia, bipolar etc.?. Did you include or not??

5. Results

A total of 977 pregnant women received ANC from March 28, 2019 to May 07, 2021. Among them, 187 (19.1%) who rejected consent were excluded and 48 (4.9%) with missing data were deleted. Here, those who refuse to take consent should be counted as non-response rate and the response rate should be clearly stated.

Table 1

• It is better to replace “Number” with “Frequency”

• It is not self-explanatory; it did not show when, where the research is conducted as stated “Table 1 characteristic of the study participants”

• Figure 1 is difficulty to understand that needs clarity for the reader.

• It is generally replace bivariate with bivariable and multivariate with “multi variable”.

Model fitness

The result of the Hosmer-Lemeshow test (P=0.034) indicated an acceptable fit of our multivariate regression model. However, it is true that a significant test indicates that the model is not a good fit and a non-significant test indicates a good fit. In this case the Hosmer-Lemeshow test is found to be significant which indicated the data is not explained by the model.

Predictor variables

You tried to assess factors that `potential affect the dependent variable. However, I noticed that important variables such as perceived stress, stressful life events, perceived stigma, substance history and history of medical illness among the participants and their families have been missed which at least would have been stated in the limitation part.

6. Discussion

Your discussion focused the comparison of your study with the previous studies but you did not address the clinical implication of your study findings. The direction of your recommendation for the scientific community is not clear.

7. Language

The manuscript would have been interesting to be reviewed with language expert.

Decision

Based on the points mentioned above, I have to decide not to consider the publication of this manuscript.

6. PLOS authors have the option to publish the peer review history of their article (what does this mean?). If published, this will include your full peer review and any attached files.

Reviewer #1: **Yes: **Dr Sara Shishehgar

Reviewer #2: No

---

## [Author Response · Author response to Decision Letter 0]

13 Jul 2023

Dear Editor and Reviewers:

Thank you for your letter and for the reviewers’ comments concerning our manuscript entitled “Prevalence and Predictors of Prenatal Depression during the COVID-19 Pandemic: A Multistage Observational Study in Beijing, China” (ID: PONE-D-22-21013). Those comments are valuable and very helpful for revising and improving our paper, as well as the important guiding significance to our research. We have studied comments carefully and have made correction which we hope meet with approval.

Suggestion and Response to academic editor

1. Please ensure that your manuscript meets PLOS ONE's style requirements, including those for file naming. The PLOS ONE style templates can be found at https://journals.plos.org/plosone/s/file?id=wjVg/PLOSOne_formatting_sample_main_body.pdf and https://journals.plos.org/plosone/s/file?id=ba62/PLOSOne_formatting_sample_title_authors_affiliations.pdf.

We have carefully studied the two documents mentioned above and revised our paper in accordance with the PLOS ONE style templates, including title, authors, affiliations, and main body. Please feel free to contact us if anything improper exists.

“This research received support from the Capital Clinical Application Research Founding of Beijing Municipal Science and Technology Committee (Grant number: Z181100001718031). The funder did not involve in any part of the study process, from design to submit the article for publication.”

Funding Statement has been updated as your suggestion and attached at the end of the cover letter 2.

3. PLOS requires an ORCID iD for the corresponding author in Editorial Manager on papers submitted after December 6th, 2016. Please ensure that you have an ORCID iD and that it is validated in Editorial Manager. To do this, go to ‘Update my Information’ (in the upper left-hand corner of the main menu), and click on the Fetch/Validate link next to the ORCID field. This will take you to the ORCID site and allow you to create a new iD or authenticate a pre-existing iD in Editorial Manager. Please see the following video for instructions on linking an ORCID iD to your Editorial Manager account: https://www.youtube.com/watch?v=_xcclfuvtxQ.

We have authenticated a pre-existing ORCID iD (0000-0002-3527-4859) in Editorial Manager.

“This research received support from the Capital Clinical Application Research Founding of Beijing Municipal Science and Technology Committee (Grant number: Z181100001718031). The funder did not involve in any part of the study process, from design to submit the article for publication.”

“This research received support from the Capital Clinical Application Research Founding of Beijing Municipal Science and Technology Committee (Grant number: Z181100001718031). The funder did not involve in any part of the study process, from design to submit the article for publication.”

Funding Statement has been amended and attached at the end of cover letter 2. Funding information in the manuscript has been removed.

Response to the Reviewer’s comments:

Reviewer #1:

1. My first concern is about the risk of bias in collecting data and reporting results in this study. Authors measured depression (interviews/survey) in different individuals in different phases of Covid 19 outbreak. We are aware that depression depends on various factors like socio-demographic characteristics, economic status, age, number of pregnancies, and mental status at the time of the study (as the authors acknowledged and discussed in the method section). However, I am still uncertain how authors reduced the risk of bias in reporting the prevalence in different phases of Covid 19 outbreak. I mean, how did they relate a higher rate of depression in certain individuals who were surveyed during the peak phase to the outbreak? For example, those who attended the study during the peak phase are not the same individuals who did so before or post the peak phase. I like to see what strategies the authors applied to reduce the risk of bias in reporting results.

It is really true as Reviewer suggested that the reduction of bias risk is very important for a multi-stage observational study, we have made an addition in Strengths and limitations section to illustrate it (Page 22, line 412-417).

Given that prenatal depression can only be measured during pregnancy and the unpredictability of COVID-19 development trajectory, longitudinal studies are almost unrealistic. This means that we were unable to recruit the same samples to conduct this study. Therefore, we assessed the homogeneity of confounding variables across three phases, such as age, BMI, education attainment, etc. Fortunately, Table S1 indicated that demographic, pregnancy-related, and psychosocial characteristics across three phases were homogenous. For this reason, we inference that a significant variation in the prevalence of prenatal depression may be partially due to the different phases of the COVID-19.

2. Another concern is about collecting data for the pre-peak phase of covid 19. How did the authors collect data? Did you conduct the study from March 2019 to May 2021, then classified the collected data into 3 phases (after completing data collection)?! Is that right? I mean, how did the authors predict there would be a peak phase for Covid 19? I would like to see an explanation about this.

We are very grateful for your concern. Our hospital was appointed as the prenatal psychological care service at November, 2018 and since then we started to collect data. Considering the duration of the peak-COVID-19, equal days, and elimination of residual effects, we intercepted data from March 28, 2019 to May 07, 2021.

Reviewer #2:

1. The body of knowledge that your study added

What was the necessity for your research, given that there have already been studies on this topic in many regions of China that have taken into account essential factors that you overlooked, and did you feel that your research contributed anything new to the body of knowledge? (I have attached some of similar works conducted previously).

I noticed that there is a duplication of efforts.

• https://www.frontiersin.org/articles/10.3389/fpsyt.2023.1191152/full

• https://bmcpregnancychildbirth.biomedcentral.com/articles/10.1186/s12884-022-04428-1

• https://bmjopen.bmj.com/content/10/9/e038511

After carefully study of literatures listed above, we have re-written introduction (Page 4, line 91-96 and page 5, line 97-106 in “Revised Manuscript with Track Changes”) and discussion (Page 20, line 356-360 in “Revised Manuscript with Track Changes”) section according to the Reviewer’s suggestion.

Firstly, the duration of these studies’ investigation was relatively short, focusing on a particular phase of the COVID-19 separately, and unable to provide a comprehensive picture of the prevalence of prenatal depression throughout the epidemic in China. By contrast, our research was a multistage observational study, and revealed the impact of the COVID-19 on prenatal depression by comparing the prevalence of prenatal depression among pre-, peak-, and post-epidemic. In addition, none of them involved gender preference in the analysis of relevant factors, which is a significant variable for prenatal depression in low- and middle-income countries, and this is exactly one highlight of our study. Therefore, we believe that this study is necessary and contributes something new.

Moreover, the data was taken from 2019-2021, and it would have been interesting if it was accessible for the scientific community during the rapid raise of the covid-19 pandemic. However, it is too late to be used by the scientific community (two years later than the time of data collection).

Just as Reviewer suggested, two years after data collection does appear to be a bit late for a study to be published. Be honest, we submitted our manuscript in June 2022, when the epidemic was still strictly controlled in Beijing. Despite the editor’s great efforts, it took nearly one year before the decision. We will do our best to make this research available to the scientific community as soon as possible.

2. Abstract

The abstract did not adhere to the PLOS ONE’s manuscript submission guidelines:

The abstract should describe the main objective(s) of the study, the methods, the results and the conclusion. The subheading “Purpose” is not included in the PLOS ONE Manuscript Submission Guidelines.

Thanks for your careful checks, we are very sorry for our incorrect writing. We have replaced the subheading “purpose” with “objective” according to submission guidelines (Page 1, line 12 in “Revised Manuscript with Track Changes”).

3. Introduction 

You stated ‘’Similarly, relatively high rates of symptoms of depression varying from 14.6% to 48.3% are reported in China, Spain, Italy, Iran, the US, Turkey, Nepal, and Denmark during the outbreak [8)’’. However, you missed the citation of all the references for studies conducted in various countries which actually indicated poor referencing for your evidences.

We are appreciative of the carefulness of the reviewers. The literature we cited is a systematic review, and the original article stated that “relatively high rates of symptoms of anxiety (6.33% to 50.9%), depression (14.6% to 48.3%), posttraumatic stress disorder (7% to 53.8%), psychological distress (34.43% to 38%), and stress (8.1% to 81.9%) are reported in the general population during the COVID-19 pandemic in China, Spain, Italy, Iran, the US, Turkey, Nepal, and Denmark”. Therefore, we have not cited references for studies in various countries, please feel free to contact us if it is necessary. 

The gaps in the research published so far have not been adequately explained, nor the significance of your study compared to previous studies. The introduction also lacks focus to the study target (prenatal depression and its predicators). Moreover, the magnitude and important predictors of prenatal depression during the pandemic are not clearly stated. Overall, it needs general improvement.

As Reviewer suggested that several potential predictors and COVID-19-related variables were important and not involved in our study. We have indicated this limitation in the “Strengths and limitations” section (Page 22, line 408-412 in “Revised Manuscript with Track Changes”). Furthermore, we have cited several similar studies in Introduction section, and explained the gaps in these studies to demonstrate the significance and implication of our work (Page 4, line 91-96 and page 5, line 97-102 in “Revised Manuscript with Track Changes”).

4. Methods

Ethical issues

The eligibility criterion lacks clarity.

Inclusion criteria for eligible were pregnant women (2-9 months of pregnancy) who were aged 20 years and older, Chinese-speaking, confirmed their written consent, and had no plans to leave residence within 1 year. Women who had life-threatening illness or severe psychiatric disorders (e.g., bipolar disorder, schizophrenia or neurodevelopmental disorder) were excluded.

• Questions

Is it ethical to exclude non -Chinese speaking pregnant mothers, and planned to leave before one year but suffering from severe prenatal depression? What about those mothers with severe depression. Because, you stated that “Women with a high probability of prenatal depression were directed to contact free professional healthcare. How did you see the deliberate exclusion of mothers with depression from health care service based on their language from the ethical perspective? Why you included mothers >20years of age. What if mothers at the age of 18, 19?

It is really true as Reviewer suggested that it is not ethical to exclude pregnant women who are non -Chinese speaking or plan to leave within 1 year. The reason for setting such inclusion criteria is that pregnant women suffering from prenatal depression in this study will be enrolled in a six-mouth psychological intervention, named “couple communication program”. Fortunately, no pregnant women were excluded due to these two criteria. We have removed these in accordance with reviewer’s suggestion (Page 6, line 121-122 in “Revised Manuscript with Track Changes”).

Mothers <20 years were excluded in the study to reduce confusion, since it is the legal age of marriage in China. Naturally, we will provide them with medical care as well.

Assessment tool

• Psychological assessment tool

You stated “Psychosocial characteristics comprised mental history and social support’’. Here, the word mental history is a vague and stigmatizing word, it is better to replace with’ history of mental illness”

Thank you for pointing this out. We have replaced the term “mental history” in the entire manuscript with “history of mental illness”.

Based on a previous study, total scores were defined as low (< 44) and high (45-66). What if the score becomes exactly 44???

We sincerely thank you for careful reading. After re-reading of the reference and confirmation by all authors, low social support was defined as ≤44 and we followed this definition in data analysis as well. We are very sorry for our mis-writing and have corrected in the manuscript (Page 9, line 202 in “Revised Manuscript with Track Changes”).

Statistical analysis

• This sentence is ambiguous for the reader.

All data were categorized variables and were presented as absolute number and percentage (%). To identify the association between prevalence and phases of COVID-19, Pearson’s Chi-square was used after the examination of confounder’s homogeneity.

We have revised this sentence for better understanding by the reader (Page 9, line 205-210 in “Revised Manuscript with Track Changes”).

All variables with a p-value of 0.2 or lower in bivariate analysis were selected as potential predictors for the multivariate logistic regression model according to the suggestion of Hosmer and Lemeshow [37], crude odds ratios (cORs) and 95% confidence intervals (CIs) were calculated.

• Question

Why only variables with a p-value <=0.2 in the bivariate analysis were taken in to multi-variate analysis. It is generally recommended to included variables known to affect the dependent variable (prenatal depression in your case) in various studies rather than taking variables with a p-value<=0.2 in the bivariable analysis.

Reviewer’s suggestion for variable inclusion is reasonable. For the same purpose, our selection for using a significance level as high as 0.2 as a screening criterion in the bivariable analysis is to prevent missing variables known to be important, just as recommendation by Hosmer and Lemeshow in Applied Logistic Regression, “Our recommendation for using a significance level as high as 0.20 or 0.25 as a screening criterion for initial variable selection is based on the work by Bendel and Afif (1977) on linear regression and on the work by Mickey and Greenland (1989) on logistic regression. These authors show that use of a more traditional level (such as 0.05) often fails to identify variables k

---

## [Editor Report · Decision Letter 1]

20 Jul 2023

PONE-D-22-21013R1

Prevalence and Predictors of Prenatal Depression during the COVID-19 Pandemic: A Multistage Observational Study in Beijing, China

PLOS ONE

Dear Dr. LIU,

Thank you for submitting your manuscript to PLOS ONE. As reviewers' comments have not fully addressed, we have decided that your manuscript does not meet our criteria for publication and must therefore be rejected.

I am sorry that we cannot be more positive on this occasion, but hope that you appreciate the reasons for this decision.

Kind regards,

Abel Fekadu Dadi, Ph.D.

Academic Editor

PLOS ONE

Additional Editor Comments:

We thank authors for their contributions; however, the reviewers comments have not been fully addressed.

---

## [Author Response · Author response to Decision Letter 1]

4 Aug 2023

Upon careful review of your suggestions, we have realized that we made some errors during the Major Revision. Listed below are details of our updated responses to the academic editor and reviewers in an effort to address all concerns. 

Comments and suggestions from the academic editor and our responses

1. Please ensure that your manuscript meets PLOS ONE's style requirements, including those for file naming. The PLOS ONE style templates can be found at https://journals.plos.org/plosone/s/file?id=wjVg/PLOSOne_formatting_sample_main_body.pdf and https://journals.plos.org/plosone/s/file?id=ba62/PLOSOne_formatting_sample_title_authors_affiliations.pdf.

After thorough learning of the two documents, we have made necessary revisions to our paper in compliance with the PLOS ONE style templates. We have updated the title, authors, affiliations, and main body. Furthermore, we have made changes to the tables and figures, and replaced supplementary material with supporting information. Please feel free to contact us if you come across any discrepancies.

“This research received support from the Capital Clinical Application Research Founding of Beijing Municipal Science and Technology Committee (Grant number: Z181100001718031). The funder did not involve in any part of the study process, from design to submit the article for publication.”

We apologize for any mistakes we may have made and any inconvenience it has caused you. We want to clarify that the Capital Clinical Application Research Founding of Beijing Municipal Science and Technology Committee (Grant number: Z181100001718031) is the sole source of support. As per your suggestion, we have updated the Funding statement as follows and attached it to the end of this letter.

This study was funded by the Capital Clinical Application Research Founding of Beijing Municipal Science and Technology Committee (https://mis.kw.beijing.gov.cn/) to QJ, grant number: Z181100001718031. The funder had no role in study design, data collection and analysis, decision to publish, or preparation of the manuscript. There was no additional external funding received for this study.

3. PLOS requires an ORCID iD for the corresponding author in Editorial Manager on papers submitted after December 6th, 2016. Please ensure that you have an ORCID iD and that it is validated in Editorial Manager. To do this, go to ‘Update my Information’ (in the upper left-hand corner of the main menu), and click on the Fetch/Validate link next to the ORCID field. This will take you to the ORCID site and allow you to create a new iD or authenticate a pre-existing iD in Editorial Manager. Please see the following video for instructions on linking an ORCID iD to your Editorial Manager account: https://www.youtube.com/watch?v=_xcclfuvtxQ.

We have successfully verified a pre-existing ORCID iD (0000-0002-3527-4859) within Editorial Manager.

“This research received support from the Capital Clinical Application Research Founding of Beijing Municipal Science and Technology Committee (Grant number: Z181100001718031). The funder did not involve in any part of the study process, from design to submit the article for publication.”

“This research received support from the Capital Clinical Application Research Founding of Beijing Municipal Science and Technology Committee (Grant number: Z181100001718031). The funder did not involve in any part of the study process, from design to submit the article for publication.”

We apologize for any inconvenience caused. We have made an amendment to the Funding statement and attached it at the end of this letter. Additionally, we have removed the funding information from the manuscript.

Comments and suggestions from the Reviewers and our responses

Reviewer #1:

1. My first concern is about the risk of bias in collecting data and reporting results in this study. Authors measured depression (interviews/survey) in different individuals in different phases of Covid 19 outbreak. We are aware that depression depends on various factors like socio-demographic characteristics, economic status, age, number of pregnancies, and mental status at the time of the study (as the authors acknowledged and discussed in the method section). However, I am still uncertain how authors reduced the risk of bias in reporting the prevalence in different phases of Covid 19 outbreak. I mean, how did they relate a higher rate of depression in certain individuals who were surveyed during the peak phase to the outbreak? For example, those who attended the study during the peak phase are not the same individuals who did so before or post the peak phase. I like to see what strategies the authors applied to reduce the risk of bias in reporting results.

As suggested by the Reviewer, reducing bias risk is crucial for a multi-stage observational study. We have added the “Strengths and limitations” section to highlight this (Page 20, line 430 and page 21, line 431-435 in “Revised Manuscript with Track Changes”).

Given the challenges posed by the uncertain development of COVID-19 and the limited window for evaluating prenatal depression during pregnancy, longitudinal studies are inherently complex. Significantly, one of the challenges encountered is the inability to recruit the same samples for research purposes. To address this limitation, we conducted an analysis of the homogeneity of confounding variables (majority of the identified relevant factors) across three phases, which encompassed demographic, psychosocial, and pregnancy-related characteristics. Our findings, as presented in S1 Table, indicated that these confounders remained consistent throughout the three phases. Consequently, we posit that significant differences, if any, in the prevalence of prenatal depression could be attributed, at least in part, to the various stages of COVID-19.

2. Another concern is about collecting data for the pre-peak phase of covid 19. How did the authors collect data? Did you conduct the study from March 2019 to May 2021, then classified the collected data into 3 phases (after completing data collection)?! Is that right? I mean, how did the authors predict there would be a peak phase for Covid 19? I would like to see an explanation about this.

Thank you for your concern. Our hospital has been appointed as a provider of prenatal psychological care service since November 2018 and we started this work since then. To ensure accuracy, we intercepted data from March 28, 2019 to May 07, 2021, taking into account the equal duration of three phases and eliminating any residual effects.

Reviewer #2:

1. The body of knowledge that your study added

What was the necessity for your research, given that there have already been studies on this topic in many regions of China that have taken into account essential factors that you overlooked, and did you feel that your research contributed anything new to the body of knowledge? (I have attached some of similar works conducted previously).

I noticed that there is a duplication of efforts.

• https://www.frontiersin.org/articles/10.3389/fpsyt.2023.1191152/full

• https://bmcpregnancychildbirth.biomedcentral.com/articles/10.1186/s12884-022-04428-1

• https://bmjopen.bmj.com/content/10/9/e038511

We have carefully learned the literature listed above and made revisions to the “Introduction” (Page 4, line 93-98 and page 5, line 99-104 in “Revised Manuscript with Track Changes”) and “Discussion” (Page 18, line 372-378 in “Revised Manuscript with Track Changes”) sections in accordance with the Reviewer’s suggestions to clarify the necessity and implication of our study.

It seems that these studies were limited in their ability to provide a comprehensive understanding of the prevalence of prenatal depression throughout the COVID-19 epidemic in China. For instance, some studies [1]lacked samples before the pandemic, while others did not compare different stages [2, 3]. By contrast, our study is unique in that it covers multiple phases of the COVID-19 epidemic and takes into account confounding variables to provide a more comprehensive picture of the prevalence of prenatal depression among pre-, peak-, and post-epidemic in China. Additionally, our study included gender preference as a significant variable, which is overlooked in these studies. We believe that our research adds new insights to the existing body of knowledge on this topic.

Moreover, the data was taken from 2019-2021, and it would have been interesting if it was accessible for the scientific community during the rapid raise of the covid-19 pandemic. However, it is too late to be used by the scientific community (two years later than the time of data collection).

As Reviewer mentioned, it may seem belated to publish a study two years after data collection. However, we submitted our manuscript in June 2022, during a time when the outbreak in Beijing was still under control. Although the editor put in an admirable effort, it took almost 1 year for a decision. We aim to release our research to the scientific community at the earliest opportunity.

2. Abstract

The abstract did not adhere to the PLOS ONE’s manuscript submission guidelines:

The abstract should describe the main objective(s) of the study, the methods, the results and the conclusion. The subheading “Purpose” is not included in the PLOS ONE Manuscript Submission Guidelines.

Thank you for reviewing our work thoroughly. We apologize for this error in our writing. As per the submission guidelines, we have replaced the subheading “Purpose” with “Objective” (Page 1, line 14 in “Revised Manuscript with Track Changes”).

3. Introduction 

You stated ‘’Similarly, relatively high rates of symptoms of depression varying from 14.6% to 48.3% are reported in China, Spain, Italy, Iran, the US, Turkey, Nepal, and Denmark during the outbreak [8)’’. However, you missed the citation of all the references for studies conducted in various countries which actually indicated poor referencing for your evidences.

Thank you for bringing up this point. The literature we cited is a systematic review, and the original article stated that “relatively high rates of symptoms of anxiety (6.33% to 50.9%), depression (14.6% to 48.3%), posttraumatic stress disorder (7% to 53.8%), psychological distress (34.43% to 38%), and stress (8.1% to 81.9%) are reported in the general population during the COVID-19 pandemic in China, Spain, Italy, Iran, the US, Turkey, Nepal, and Denmark”. We understand that you may require additional references, please do not hesitate to let us know if it is necessary. 

The gaps in the research published so far have not been adequately explained, nor the significance of your study compared to previous studies. The introduction also lacks focus to the study target (prenatal depression and its predicators). Moreover, the magnitude and important predictors of prenatal depression during the pandemic are not clearly stated. Overall, it needs general improvement.

Thank you very much for your critical and constructive comments, it is very helpful in improving our writing. We have taken note of your suggestions and have added and enriched the “Introduction” section to better explain the gaps in previous studies and the significance and implications of our work (Page 4, line 93-98 and page 5, line 99-104 in “Revised Manuscript with Track Changes”). Additionally, we have provided further explanation for our study’s selection of potential demographic, pregnancy-related, and psychosocial predictors (Page 5, line 105-116 in “Revised Manuscript with Track Changes”). If you have any further comments or questions, please do not hesitate to reach out to us.

We have acknowledged the Reviewer’s suggestion that certain predictors and COVID-19-related variables were significant and should have been included in our study. This limitation has been stated in the “Strengths and limitations” section (Page 20, line 426-430 in “Revised Manuscript with Track Changes”). Our team will strive to explore and enhance this aspect in future research.

4. Methods

Ethical issues

The eligibility criterion lacks clarity.

Inclusion criteria for eligible were pregnant women (2-9 months of pregnancy) who were aged 20 years and older, Chinese-speaking, confirmed their written consent, and had no plans to leave residence within 1 year. Women who had life-threatening illness or severe psychiatric disorders (e.g., bipolar disorder, schizophrenia or neurodevelopmental disorder) were excluded.

• Questions

Is it ethical to exclude non -Chinese speaking pregnant mothers, and planned to leave before one year but suffering from severe prenatal depression? What about those mothers with severe depression. Because, you stated that “Women with a high probability of prenatal depression were directed to contact free professional healthcare. How did you see the deliberate exclusion of mothers with depression from health care service based on their language from the ethical perspective? Why you included mothers >20years of age. What if mothers at the age of 18, 19?

We apologize for the unclear description in the paper. As suggested by the Reviewer, it is not ethical to exclude pregnant women who do not speak Chinese or plan to leave within 1 year. We excluded non- Chinese speaking pregnant women to avoid any bias due to language or written misunderstanding. Additionally, pregnant women suffering from prenatal depression were further enrolled in a six-month psychological intervention called the “couple communication program”, whereby pregnant women who planned to leave within 1 year were excluded. However, we are still committed to providing professional healthcare to all pregnant women, regardless of inclusion or not. Given that no pregnant women were excluded because of “no plans to leave residence within 1 year”, we have removed this (Page 6, line 132 in “Revised Manuscript with Track Changes”) and added a clarification on how to manage the high-risk pregnant women (Page 6, line 144-145 in “Revised Manuscript with Track Changes”). We excluded mothers <20 years to avoid confusion since the legal age of marriage in China is 20. Naturally, we will provide them with psychological healthcare as well. As a designated prenatal psychological care service provider, it is our duty to ensure all pregnant women receive the care they need.

Assessment tool

• Psychological assessment tool

You stated “Psychosocial characteristics comprised mental history and social support’’. Here, the word mental history is a vague and stigmatizing word, it is better to replace with’ history of mental illness”

Thank you for bringing this to our attention. We have made the necessary correction throughout the entire manuscript by replacing the term “mental history” with “history of mental illness”.

Based on a previous study, total scores were defined as low (< 44) and high (45-66). What if the score becomes exactly 44???

We sincerely thank you for your careful reading. A

---

## [Decision Letter · Decision Letter 2]

13 Dec 2023

PONE-D-22-21013R2

Prevalence and predictors of prenatal depression during the COVID-19 pandemic: a multistage observational study in Beijing, China

PLOS ONE

Dear Dr. LIU,

Thank you for submitting your manuscript to PLOS ONE. After careful consideration, we feel that it has merit but does not fully meet PLOS ONE’s publication criteria as it currently stands. Therefore, we invite you to submit a revised version of the manuscript that addresses the points raised during the review process.

We look forward to receiving your revised manuscript.

Kind regards,

Abera Mersha, MSc.

Academic Editor

PLOS ONE

Journal Requirements:

Additional Editor Comments (if provided):

Reviewers' comments:

Reviewer's Responses to Questions

**Comments to the Author**

1. If the authors have adequately addressed your comments raised in a previous round of review and you feel that this manuscript is now acceptable for publication, you may indicate that here to bypass the “Comments to the Author” section, enter your conflict of interest statement in the “Confidential to Editor” section, and submit your "Accept" recommendation.

Reviewer #3: All comments have been addressed

Reviewer #4: All comments have been addressed

2. Is the manuscript technically sound, and do the data support the conclusions?

Reviewer #3: Yes

Reviewer #4: Yes

3. Has the statistical analysis been performed appropriately and rigorously? 

Reviewer #3: Yes

Reviewer #4: Yes

4. Have the authors made all data underlying the findings in their manuscript fully available?

Reviewer #3: Yes

Reviewer #4: No

5. Is the manuscript presented in an intelligible fashion and written in standard English?

Reviewer #3: Yes

Reviewer #4: Yes

6. Review Comments to the Author

Reviewer #3: I commend the authors for a good job in contributing to the pool of knowledge on matters maternal mental health.

The previous reviewers have been thorough in addressing the concerns on this paper.

My concerns are as follows:

1. Abstract

The authors have used the term "loose" to describe the quarantine measures put in place in other areas and "strictest" to describe the measures put in place in Beijing. I recommend change of terminology from "loose" to "lax" or any other scientific inclined terminology and from "strictest" to "strict" or "stringent". This is a suggestion and the authors may choose to incorporate it or not.

2. Introduction

Definition of terms: Most key terms were not introduced to the reader by clearly defining them earlier in the paper e.g ante-natal depression, quarantine, COVID-19 etc. The author should define these terms as he/she gives some background for ease of understanding this work especially if you are interacting with these terms for the first time while reading this paper. I propose that they pick definitions from more acceptable sources e.g WHO.

3. Methodology

Study site: Did the choice of a tertiary institution in the heart of a big city do justice to "sexual preference" variable? Looking at the majority of the respondents, close to 90% had at least a university degree and 48% were primigravida (to mean 1st pregnancy). My concern is on the sex preference response bias, to a highly educated mother , the sex of the child may not be a big concern compared to the less educated mother in the rural areas. Secondly, a first time mother is rarely keen on sex of the child unless her intention is to get just one child compared to a second time mother who may be keen on getting a different sex from the first child.

4. Findings and Conclusion

This paper suggests that social support is paramount in supporting the mental health of pregnant mothers while the conclusion applauds the strict quarantine measures Beijing had put in place. The authors are contradicting themselves to a certain extent, we know social activities which are great therapy and acts as a preventive measures for mental illness were affected by the stringent measures during the pandemic especially in low and middle income countries which thrive on social activities. They need to be elaborative on how each is important in improving maternal mental health e.g stringent measures will help in reducing anxiety of contracting infectious diseases and social support can be a preventive or therapeutic measure.

I want to appreciate the team for putting this together and the editor for allowing me to review this work. I look forward to interacting with this manuscript as a published paper after the minor revision. This paper, once published, will act as a guide in future pandemics on how to manage the mental health component of special groups.

Should there be any concern, I will be glad to address them.

Reviewer #4: Under abstract, where sentence on objective starting with "this study aimed to compare the prevalence of prenatal depression among??.... The authors to indicate whether they intentionally omitted "pregnant women".

The same omission or otherwise stated has been observed under objective line 17.

Still under abstract and throughout the text, "boypreference" has been portrayed to be one word.

Also, conclusion line 37 should read " not significant" and not " Insignificant".

The authors did not attach data collection tool as well as the dataset.

For line 280-294, all the "p" in p- values should be in lower case.

The authors have in general done a commendable job and have also adequately responded to previous reviewer's comments.

7. PLOS authors have the option to publish the peer review history of their article (what does this mean?). If published, this will include your full peer review and any attached files.

Reviewer #3: No

Reviewer #4: No

---

## [Author Response · Author response to Decision Letter 2]

22 Dec 2023

Listed below are details of our point-to-point responses to the Academic Editor and reviewers to address all concerns. Your suggestions and comments were presented in italics and our responses were in red.

Journal Requirements:

Based on requests from the Academic Editor, we have carefully reviewed our references and made some modifications to ensure completeness and correctness. Details are as follows:

 1. After careful checking, we ensured that no retracted literature was cited in our manuscript.

2. To enhance readability, we have replaced the abbreviations of the two institutions with their full names, replacing WHO with World Health Organization (Page 22, line 466 in “Revised Manuscript with Track Changes 3”, NO. 5) and NHC with Chinese National Health Commission (Page 30, line 667 in “Revised Manuscript with Track Changes 3”, No. 64) respectively.

3. We updated the year, volume, issue, and pages of 2 electronic publication papers (Page 23, line 494-497 in “Revised Manuscript with Track Changes 3”, No. 13; Page 31, line 686-689 in “Revised Manuscript with Track Changes 3”, No. 69). 

4. Based on the recommendation of Reviewer #3, we cited 2 references to define prenatal depression and quarantine (Page 23, line 480-481 in “Revised Manuscript with Track Changes 3”, No. 9; Page 24, line 514-515 in “Revised Manuscript with Track Changes 3”, No. 19).

5. “Nasiripour AA, Kazemi MAA, Izadi A. Designing a hospital performance assessment model based on balanced scorecard. Healthmed. 2012;6(9):2983-2989” is a fault citation, and we replaced with “Su Y, D'Arcy C, Meng X. Research Review: Developmental origins of depression - a systematic review and meta‐analysis. J CHILD PSYCHOL PSYC. 2020;62(9):1050-66. doi: 10.1111/jcpp.13358.” to clarify the role of socioeconomic status in the development of depression (Page 29, line 640-642 in “Revised Manuscript with Track Changes 3”, No. 56).

Comments and suggestions from the Reviewers and our responses

Reviewer #3:

1.Abstract

The authors have used the term "loose" to describe the quarantine measures put in place in other areas and "strictest" to describe the measures put in place in Beijing. I recommend change of terminology from "loose" to "lax" or any other scientific inclined terminology and from "strictest" to "strict" or "stringent". This is a suggestion and the authors may choose to incorporate it or not.

We would like to express our gratitude for your excellent recommendation, this will be helpful to improve the scientific and rigorous of our paper. We have made appropriate changes based on your advice (Page 1, line 16 and Page 5, line 97 in “Revised Manuscript with Track Changes 3”; Page 1, line 18; Page 5, line 120; and Page 18, line 364 in “Revised Manuscript with Track Changes 3”).

2. Introduction

Definition of terms: Most key terms were not introduced to the reader by clearly defining them earlier in the paper e.g ante-natal depression, quarantine, COVID-19 etc. The author should define these terms as he/she gives some background for ease of understanding this work especially if you are interacting with these terms for the first time while reading this paper. I propose that they pick definitions from more acceptable sources e.g WHO.

Based on your suggestion, we have added the definition of prenatal depression and quarantine to improve readability. Prenatal depression is defined as a major depressive episode during pregnancy according to the DSM-5 (Page 3, line 60-63 in “Revised Manuscript with Track Changes 3”). Quarantine is defined as an effective strategy to contain the spread of infectious diseases, but it may cause adverse psychological effects (Page 4, line 84-86 in “Revised Manuscript with Track Changes 3”). Additionally, in light of the fact that the development of COVID-19 and the statement of the WHO are described at the beginning of the Introduction, we did not modify the definition of COVID-19. If it is necessary, please do not hesitate to reach out to us.

3. Methodology

Study site: Did the choice of a tertiary institution in the heart of a big city do justice to "sexual preference" variable? Looking at the majority of the respondents, close to 90% had at least a university degree and 48% were primigravida (to mean 1st pregnancy). My concern is on the sex preference response bias, to a highly educated mother, the sex of the child may not be a big concern compared to the less educated mother in the rural areas. Secondly, a first time mother is rarely keen on sex of the child unless her intention is to get just one child compared to a second time mother who may be keen on getting a different sex from the first child.

We thank you for this comment. We fully agree with your concerns about sex preference response bias. As your statement, our samples with higher socioeconomic status and education may underestimate the sex preference, and we will endeavor to further explore the role of sex preference in the development of prenatal depression with a nationally representative sample in future research. However, we suggest that regardless of the presence of sex preference response bias, it has no impact on our conclusion that sex preference increases the risk of prenatal depression in the Haidian district, Beijing.

4. Findings and Conclusion

This paper suggests that social support is paramount in supporting the mental health of pregnant mothers while the conclusion applauds the strict quarantine measures Beijing had put in place. The authors are contradicting themselves to a certain extent, we know social activities which are great therapy and acts as a preventive measures for mental illness were affected by the stringent measures during the pandemic especially in low and middle income countries which thrive on social activities. They need to be elaborative on how each is important in improving maternal mental health e.g stringent measures will help in reducing anxiety of contracting infectious diseases and social support can be a preventive or therapeutic measure.

Thank you very much for your constructive suggestion. In the Introduction section, we clarified the gaps between the influence of different quarantine measures on maternal mental health in the available studies to introduce our study purpose. Namely, stringent measures would increase the risk of psychological illness by limiting socialization, but it is also helpful in reducing the fear of infection in pregnant women (Page 4, line 83-94 in “Revised Manuscript with Track Changes 3”). Unfortunately, as you mentioned, we failed to provide adequate echoes in the Conclusion section. Indeed, by analyzing the variation in prenatal depression rates across 3 phases of COVID-19, we concluded that, on the whole, stringent measures appear to be protective of the mental health of pregnant women during the epidemic. We have made additions to the Conclusion section to illustrate this point (Page 18, line 364 in “Revised Manuscript with Track Changes 3”). 

In comparison, our conclusions about social support were based on the overall data and did not analyze changes in levels of social support at different stages of COVID-19. We will attempt to examine this and explore how to provide adequate social support in the context of stringent quarantine measures in future works.

Reviewer #4:

Under abstract, where sentence on objective starting with "this study aimed to compare the prevalence of prenatal depression among??.... The authors to indicate whether they intentionally omitted "pregnant women".

The same omission or otherwise stated has been observed under objective line 17.

Thank you very much for your careful review. We indicated that in light of the limitation of 300 words, we intentionally shortened our Abstract. If it is essential to specify the study population (pregnant women) even though prenatal depression has already been described, please feel free to contact us.

Still under abstract and throughout the text, "boypreference" has been portrayed to be one word.

We have carefully reread our paper and did not detect the word “boypreference”. We are not sure if you mean that “boy-preference” is not the right description.

Also, conclusion line 37 should read " not significant" and not " Insignificant".

Thank you for pointing this out. We have corrected it as per your request (Page 2, line 37 and Page 21, line 429-430 in “Revised Manuscript with Track Changes 3”).

The authors did not attach data collection tool as well as the dataset.

We are very sorry for our omission to upload the original dataset underlying the findings and we have uploaded it as supporting information during this revision.

For line 280-294, all the "p" in p- values should be in lower case.

We have replaced all the “P” with “p” throughout the paper. We are impressed by your rigorous research style.

---

## [Decision Letter · Decision Letter 3]

18 Jan 2024

PONE-D-22-21013R3Prevalence and predictors of prenatal depression during the COVID-19 pandemic: a multistage observational study in Beijing, ChinaPLOS ONE

Dear Dr. LIU,

Thank you for submitting your manuscript to PLOS ONE. After careful consideration, we feel that it has merit but does not fully meet PLOS ONE’s publication criteria as it currently stands. Therefore, we invite you to submit a revised version of the manuscript that addresses the points raised during the review process.

We look forward to receiving your revised manuscript.

Kind regards,

Abera Mersha, MSc.

Academic Editor

PLOS ONE

Journal Requirements:

Reviewers' comments:

Reviewer's Responses to Questions

**Comments to the Author**

1. If the authors have adequately addressed your comments raised in a previous round of review and you feel that this manuscript is now acceptable for publication, you may indicate that here to bypass the “Comments to the Author” section, enter your conflict of interest statement in the “Confidential to Editor” section, and submit your "Accept" recommendation.

Reviewer #3: All comments have been addressed

Reviewer #4: All comments have been addressed

2. Is the manuscript technically sound, and do the data support the conclusions?

Reviewer #3: Yes

Reviewer #4: Yes

3. Has the statistical analysis been performed appropriately and rigorously? 

Reviewer #3: Yes

Reviewer #4: Yes

4. Have the authors made all data underlying the findings in their manuscript fully available?

Reviewer #3: Yes

Reviewer #4: Yes

5. Is the manuscript presented in an intelligible fashion and written in standard English?

Reviewer #3: Yes

Reviewer #4: Yes

6. Review Comments to the Author

Reviewer #3: The concerns I raised have been well addressed. Thank you and I wish you the best in the next phase of this manuscript submission.

Reviewer #4: Table 1, line 250: All variables such as age, education attainment should begin with capital letters

Line 296 (p values>0.05) should be presented as p>0.05

Conclusion (line 425) "We concluded that the public and professionals should take prenatal depression seriously during the COVID-19" should read "We concluded that the public and professionals should take prenatal depression seriously during the COVID-19"

Reference 25, the journal title "FRONT PSYCHIATRY" should capitalize first letters of the word as "Front Psychiatry".

Similarly line 559 reference 32 "LANCET GLOBAL HEALTH" should be written as "Lancet Global Health".

Line 693 reference 71 "MULTIDISCIPLINARY HEALTH" to be written as "Multidisciplinary Health".

The image quality of figure 2 and figure 3 is not very clear to enable reading the data output easily. Consider rendering these images (figures) afresh to improve on their quality.

The manuscript can be considered for publication once the corrections have been made, there is therefore no need for another round of resubmission to me as a reviewer.

Thank you.

7. PLOS authors have the option to publish the peer review history of their article (what does this mean?). If published, this will include your full peer review and any attached files.

Reviewer #3: No

Reviewer #4: No

---

## [Author Response · Author response to Decision Letter 3]

18 Jan 2024

Comments and suggestions from the Reviewers and our responses

Reviewer #3:

The concerns I raised have been well addressed. Thank you and I wish you the best in the next phase of this manuscript submission.

We are impressed by your friendly and positive comments throughout the review process, please accept our sincerest thanks and best wishes.

Reviewer #4: 

Table 1, line 250: All variables such as age, education attainment should begin with capital letters.

We have amended them in Table 1 and thank you very much for your suggestion (Page 11, line 250 in “Revised Manuscript with Track Changes 4”).

Line 296 (p values>0.05) should be presented as p>0.05.

We have amended it and thank you very much for your suggestion (Page 15, line 296 in “Revised Manuscript with Track Changes 4”).

Conclusion (line 425) "We concluded that the public and professionals should take prenatal depression seriously during the COVID-19" should read "We concluded that the public and professionals should take prenatal depression seriously during the COVID-19"

We are very sorry for the confusion in your description. It seems that your statement does not differ from our paper. Please do not hesitate to point out the inappropriateness of our sentence clearly.

Reference 25, the journal title "FRONT PSYCHIATRY" should capitalize first letters of the word as "Front Psychiatry".

Similarly line 559 reference 32 "LANCET GLOBAL HEALTH" should be written as "Lancet Global Health".

Line 693 reference 71 "MULTIDISCIPLINARY HEALTH" to be written as "Multidisciplinary Health".

We are very impressed by your careful review. We have modified the abbreviations of the journals’ names as per your recommendation and will keep this in mind in future writing.

The image quality of figure 2 and figure 3 is not very clear to enable reading the data output easily. Consider rendering these images (figures) afresh to improve on their quality.

We have rendered and uploaded them. We are impressed by your rigorous research style.

---

## [Editor Report · Decision Letter 4]

23 Jan 2024

Prevalence and predictors of prenatal depression during the COVID-19 pandemic: a multistage observational study in Beijing, China

PONE-D-22-21013R4

Dear Dr. LIU,

We’re pleased to inform you that your manuscript has been judged scientifically suitable for publication and will be formally accepted for publication once it meets all outstanding technical requirements.

Kind regards,

Abera Mersha, MSc.

Academic Editor

PLOS ONE